# The Formation of Mn-Ce-Zr Oxide Catalysts for CO and Propane Oxidation: The Role of Element Content Ratio

Tatyana N. Afonasenko [1], Daria V. Yurpalova [1], Zakhar S. Vinokurov [2,3], Andrey A. Saraev [2,3], Egor E. Aidakov [2], Valeriya P. Konovalova [2], Vladimir A. Rogov [2] and Olga A. Bulavchenko [2,*]

[1] Center of New Chemical Technologies BIC, Boreskov Institute of Catalysis, Neftezavodskaya St., 54, 644040 Omsk, Russia
[2] Boreskov Institute of Catalysis SB RAS, Lavrentiev Ave. 5, 630090 Novosibirsk, Russia
[3] Synchrotron Radiation Facility SKIF, Boreskov Institute of Catalysis SB RAS, Nikol'skiy Prospekt 1, 630559 Kol'tsovo, Russia
* Correspondence: obulavchenko@catalysis.ru

**Abstract:** The $MnO_x$-$ZrO_2$-$CeO_2$ oxide catalysts were synthesized by co-precipitation method with varying (1) Zr/Zr + Ce molar ratio at constant manganese content of 0.3; (2) manganese content at constant Zr/Ce molar ratio of 1; (3) Mn/Mn + Zr molar ratio at constant Ce content of 0.5. Catalysts are characterized by XRD, $N_2$ adsorption, TPR, and XPS. The catalytic activity of all the series was tested in the CO and propane oxidation reactions. In contrast to the variation of the manganese content, the Zr/Zr + Ce molar ratio does not significantly affect the catalytic properties. The dependence of the catalytic activity in CO oxidation on the manganese content has a «volcano» shape, and the best catalytic performance is exhibited by the catalyst with Mn/(Zr + Ce) = 1. In the case of propane oxidation reaction, there is «sigma» like dependence, activity increases with increase of Mn/(Mn + Zr + Ce) molar ratio up to 0.3, stabilizing with a further increase in the manganese content. XRD and XPS have shown that with an increase of the Mn concentration in the $MnO_x$-$ZrO_2$-$CeO_2$ catalysts, the amount of crystalline manganese oxides such as $Mn_2O_3$ and $Mn_3O_4$, as well as the surface concentration of Mn cations, increases. While the content of $Mn_xZr_yCe_{1-x-y}O_2$ solid solution decreases, the concentration of manganese cations (x) in volume of $Mn_xZr_yCe_{1-x-y}O_2$ mixed oxide grows. The maximum activity in CO oxidation corresponds to the balance between the amount of the solid solution and the concentration of manganese cations in the volume of mixed oxide. The propane oxidation reaction is less sensitive to the state of manganese ion rather than to its amount. In this case, a decrease in the content of the $Mn_xZr_yCe_{1-x-y}O_2$ solid solution with increase in manganese amount in catalyst is compensated by an increase in content of crystalline manganese oxides and the surface concentration of manganese.

**Keywords:** $Mn_xZr_yCe_{1-x-y}O_2$ solid solution; CO oxidation; propane oxidation

## 1. Introduction

Transition metal oxide catalysts have been widely accepted as promising catalysts for the combustion of CO, hydrocarbons, and volatile organic compounds. Among these, manganese oxide catalysts are attracting a great deal of attention since they are cheap, active, and environment-friendly. Generally, the activity of manganese catalysts correlates with their redox properties [1–3]. Manganese ions easily change the oxidation state, forming various types of oxides such as $MnO_2$, $Mn_5O_8$, $Mn_2O_3$, $Mn_3O_4$, and MnO, and provide highly mobile oxygen for oxidation reactions [4].

Doping the cerium oxides with manganese significantly increases catalytic activity compared to bulk manganese oxides [5]. Due to the easy transition between $Ce^{3+}$ and $Ce^{4+}$ and the high mobility of $O^{2-}$ ions in the $CeO_2$ lattice, ceria modifies the redox activity of manganese, stabilizing it in higher oxidation states at low temperatures that is preferable

for oxidation reactions and facilitating oxygen migration [6]. Apart from the formation of labile oxygen and promotion of the process of oxygen activation and transfer, ceria has a significant oxygen storage capacity and is itself active in oxidative reactions, for example, in soot oxidation [7]. The inclusion of zirconium in the ceria lattice improves the thermal stability of oxide and further increases the oxygen mobility [6,8]. Terribille et al. [6] found that the highest reduction degree and the lowest reduction temperatures are achieved at an equimolar ratio of cerium and zirconium $Ce_{0.5}Zr_{0.5}O_2$.

The properties of ternary Mn-Zr-Ce catalysts have been extensively studied in the reactions of total oxidation of O-, Cl-containing volatile organic substances and CO [9–13], soot oxidation [14,15], and removal of $NO_x$ by selective catalytic reduction with ammonia [16–18]. Catalytic properties of Mn–Zr–Ce oxides depend on the preparation procedure and the ratio between elements. These factors determine the state of the active component: supported $MnO_x$ nanoparticles or defective Mn-Zr-Ce solid solution. Depending on the preparation conditions, one of these states prevails. In the case of impregnation methods, manganese nanoparticles are spread uniformly on the surface of support. The impregnation approach effectiveness was shown for a low content of manganese. Kaplin et al. compared 8 wt% $MnO_x$-$Ce_{0.8}Zr_{0.2}O_2$ prepared by co-precipitation and wet impregnation and showed that in the case of impregnation no manganese ions are embedded into the crystal lattice of $Ce_{0.8}Zr_{0.2}O_2$ oxide, while the surface of catalyst contains $MnO_x$ states [12]. Hou et al. utilized the $Ce_{0.65}Zr_{0.35}O_2$ as support for $MnO_x$/$Ce_{0.65}Zr_{0.35}O_2$ catalysts for oxidation of toluene, and the catalyst with 15 wt% $MnO_x$ loading performed the best [13]. However, at relatively high manganese content, the formation of a solid solution has a greater impact on the catalysis properties than $MnO_x$ nanoparticles. Solution combustion, co-precipitation, and sol–gel methods were used to prepare a predominantly $Mn_xZr_yCe_{1-x-y}O_2$ solid solution. Long et al. developed a Mn-Ce-Zr ternary mixed oxides catalyst by solution combustion method [19]. Zhu et al. synthesized Mn-Ce-Zr catalysts by the sol–gel method varying the molar ratio of Mn/(Mn + Ce + Zr), while the molar ratio of Ce/Zr was maintained as 1, the $Mn_{0.67}Ce_{0.16}Zr_{0.16}$ catalyst showed maximum activity in the series [11]. The $Zr_{0.4}Ce_{0.6-x}Mn_xO_2$ solid solutions were synthesized by a sol–gel method and tested in the total oxidation of butanol. The presence of both $Ce^{4+}$/$Ce^{3+}$ and $Mn^{4+}$/$Mn^{3+}$ redox couples led to excellent catalytic activity for n-butanol complete oxidation with an activity maximum achieved for $Zr_{0.4}Ce_{0.24}Mn_{0.36}O_2$ [9].

Most studies of Mn-Zr-Ce catalysts are aimed at determining the effect of manganese content, method of preparation, and support composition on the catalytic activity. To the best of our knowledge, no systematic studies on the role of element ratio have been reported for Mn-Zr-Ce catalysts synthetized by co-precipitation method, although the composition of the catalyst is an important factor determining catalytic activity. Understanding how the composition of the catalyst affects the state of the active component and its properties in the target reaction is the key to successfully achieving the best performance of the catalytic process [20,21]. In this work, we studied the effect of composition on the catalytic, structural, microstructural, and redox properties of the Mn-Ce-Zr oxide catalysts. Three series of Mn-Ce-Zr catalysts were synthesized by co-precipitation with varying (1) Zr/Zr + Ce molar ratio at constant manganese content of 0.3; (2) manganese content at constant Zr/Ce molar ratio of 1; (3) Mn/Mn + Zr molar ratio at constant Ce content of 0.5. The physicochemical properties were studied by X-ray diffraction (XRD), temperature-programmed reduction (TPR-$H_2$), and X-ray photoelectron spectroscopy (XPS), the catalytic activity was tested in the reaction of CO and propane oxidation.

## 2. Results

### 2.1. Variation of Mn/(Mn + Zr) Molar Ratio

The influence of the Mn/(Mn + Zr) molar ratio on the catalytic properties was studied for the series of (Mn,Zr)0.5Ce0.5 catalysts (Mn/(Mn + Zr) molar ratio was in the range from 0 to 1 at constant Ce content of 0.5). As reference, the Zr0.5Ce0.5 sample was used. Figure 1a,c show the temperature dependence of CO and $C_3H_8$ conversion a for

(Mn,Zr)0.5Ce0.5 catalysts. Figure 1b,d illustrate the temperature of 50% conversion of CO and $C_3H_8$ depending on the Mn/(Mn + Zr) molar ratio. For the Zr0.5Ce0.5 catalyst, CO oxidation begins at a reaction temperature of 250 °C, and 50% CO conversion is achieved at 389 °C. For (Mn,Zr)0.5Ce0.5 samples, CO conversion curves drastically shift to lower temperatures with increasing Mn content. In the range of Mn/(Mn + Zr) from 0.2 to 0.8, a noticeable shift of the CO conversion curve to a lower temperature region was observed. The $T_{50}$ gradually decreases from 179 to 157 °C for the Mn/(Mn + Zr) ratio of 0.2 and 0.8, correspondingly. Sample Mn0.5Ce0.5 activity does not change significantly, and $T_{50}$ is 160 °C, which is slightly higher than for Mn0.4Zr0.1Ce0.5.

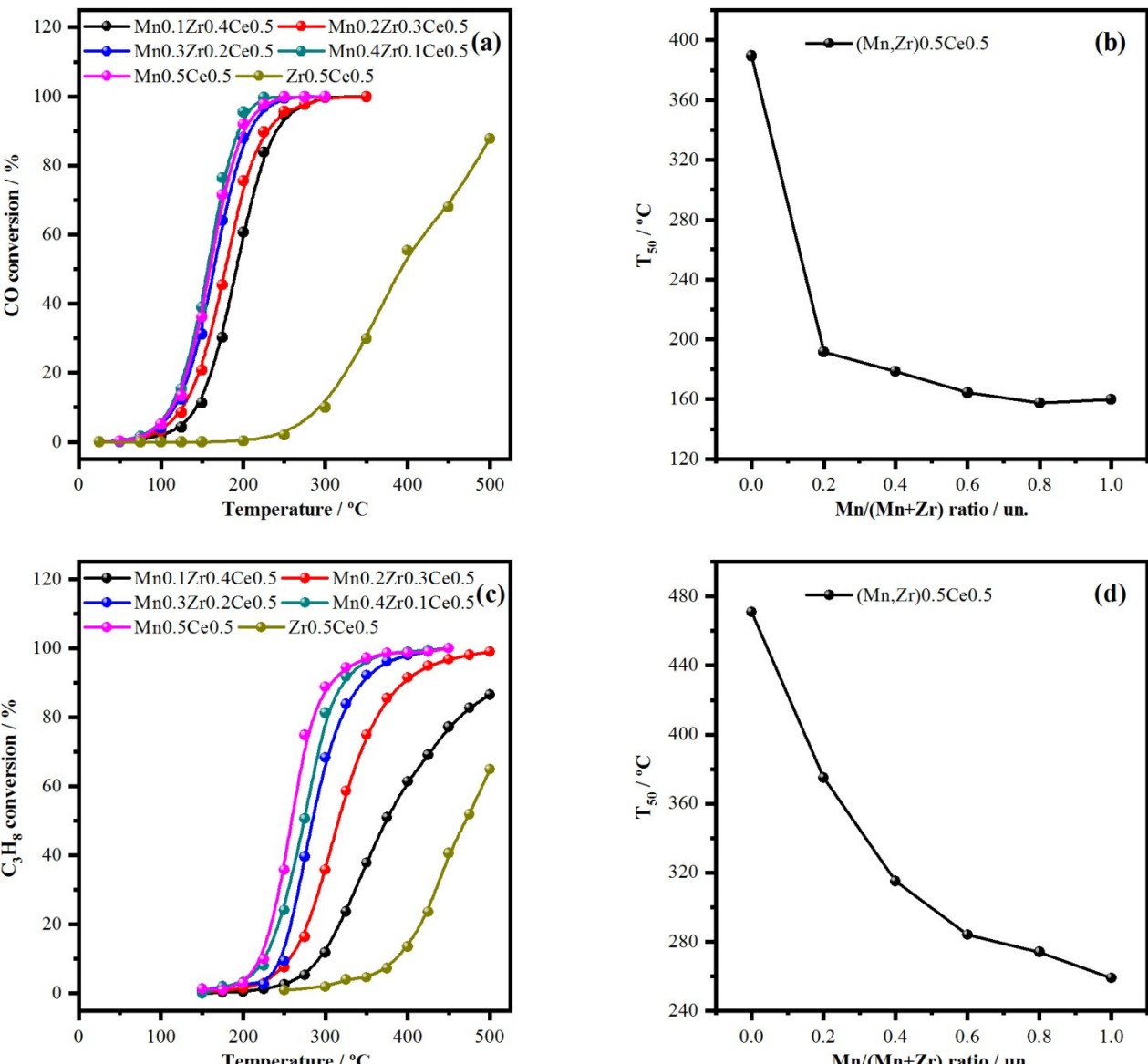

**Figure 1.** Reaction temperature dependence of (**a**) CO and (**c**) $C_3H_8$ conversion; the molar ratio Mn/(Mn + Zr) dependence of $T_{50}$ for (**b**) CO and (**d**) $C_3H_8$ conversion in oxidation reaction over (Mn,Zr)0.5Ce0.5 catalysts.

In the reaction of propane oxidation, a smoother change in the behavior of the catalyst is observed. The curves of the propane conversion gradually shift to lower temperatures with increasing Mn content, $T_{50}$ for the Zr0.5Ce0.5 sample is 470 °C and drops to 260 °C for the sample Mn0.5Ce0.5 (Mn/(Mn + Zr) = 1).

Figure 2 shows XRD patterns for (Mn,Zr)0.5Ce0.5 catalysts. All XRD patterns contain wide peaks located at 2θ = 13.1, 15.1, 21.4, 25.1, 26.3, 30.4, 33.3, and 34.2. These peaks correspond to the $CeO_2$ fluorite-type structure (sp.gr. Fm(-)3m, PDF No. 43-1002). The lattice parameter *a* of fluorite-type phase in the sample Ce0.5Mn0.5 is equal to 5.397 Å (Table 1). This value slightly differs from the lattice parameter of pure $CeO_2$ (5.411 Å), indicating the formation of $Ce(Mn)O_2$ solid solution, since the ionic radii of $Ce^{4+}$ and $Mn^{3+}$ are 0.97 Å and 0.66 Å, respectively [22]. For the samples with high Mn content, additional reflections were observed belonging to $Mn_3O_4$ (sp.gr. $I4_1$/amd, PDF No. 24-0734) and $Mn_2O_3$ (sp.gr. Ia(-)3, PDF No. 41-1442). An increase in the content of Zr results in a decrease of the amount of crystalline Mn oxides and appearance of the oxide based on the structure of tetragonal $ZrO_2$, as evidenced by the appearance of additional reflections at 2θ = 13.4, 15.4, 15.6, 22.0, and 22.1, corresponding to 011, 002, 110, 112, and 020 reflections ($ZrO_2$, sp.gr. $P4_2$/nmc, PDF No. 42-1164). For a convenient comparison of the lattice parameters of mixed oxides, we used the normalized lattice parameter of the unit cell in the case of tetragonal oxide (Table 1, a * = (a × a × c)$^{1/3}$). The normalized lattice parameter of oxide with tetragonal structure varies in the range of 5.213–5.354 Å, which significantly differs from the value of pure $ZrO_2$ (a * = 4.110 Å, PDF No. 42-1164), indicating the formation of the second solid solution t-$Zr(Ce,Mn)O_2$. Unfortunately, for both solid solutions it is impossible to estimate its composition from the cell parameters. The variation of Mn/Mn + Zr molar ratio leads to a change in the lattice parameters of mixed oxides and their content ratio, indicating the redistribution of cations between two oxides $Ce(Mn,Zr)O_2$ and t-$Zr(Ce,Mn)O_2$ (Table 1). CSR of solid solutions changes in the range of 50–130 Å.

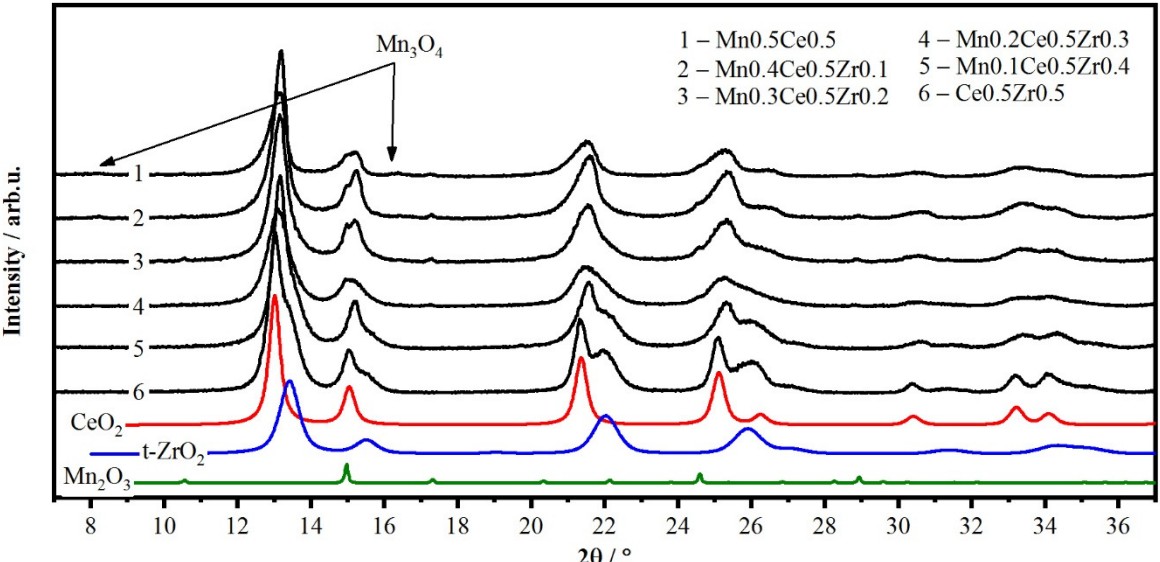

**Figure 2.** XRD patterns of (Mn,Zr)0.5Ce0.5 catalysts and peak positions of phases $CeO_2$ (PDF No. 431002), t-$ZrO_2$ (PDF No. 42-1164), $Mn_2O_3$ (PDF No. 41-1442), and $Mn_3O_4$ (PDF No. 24-0734).

**Table 1.** Phase composition, lattice parameter of solid solution (Ce(Mn,Zr)$O_2$, and t-Zr(Ce,Mn)$O_2$CSR and surface area of (Mn,Zr)0.5Ce0.5 catalysts.

| Catalyst | Phase Content, % Wt. | Lattice Parameter *, Å | CSR, Å | $S_{BET}$, m$^2$/g |
|---|---|---|---|---|
| Mn0.1Ce0.5Zr0.4 | 60% Ce(Mn,Zr)$O_2$<br>40% t-Zr(Ce,Mn)$O_2$ | 5.371<br>5.213 | 100<br>70 | 87 |
| Mn0.2Ce0.5Zr0.3 | 46% Ce(Mn,Zr)$O_2$<br>52% t-Zr(Ce,Mn)$O_2$<br>2% Mn$_2$O$_3$ | 5.391<br>5.297<br>- | 70<br>40<br>230 | 103 |
| Mn0.3Ce0.5Zr0.2 | 53% Ce(Mn,Zr)$O_2$<br>43% t-Zr(Ce,Mn)$O_2$<br>4% Mn$_2$O$_3$ | 5.383<br>5.307<br>- | 80<br>30<br>270 | 91 |
| Mn0.4Ce0.5Zr0.1 | 36% Ce(Mn,Zr)$O_2$<br>56% t-Zr(Ce,Mn)$O_2$<br>6% Mn$_2$O$_3$<br>2% Mn$_3$O$_4$ | 5.414<br>5.354<br>-<br>- | 50<br>130<br>110<br>80 | 77 |
| Ce0.5Zr0.5 | 40% (Ce,Zr)$O_2$<br>60% t-(Zr,Ce)$O_2$ | 5.396<br>5.241 | 120<br>60 | 66.5 |
| Mn0.5Ce0.5 | 87% Ce(Mn)$O_2$<br>8% Mn$_2$O$_3$<br>5% Mn$_3$O$_4$ | 5.397<br>-<br>- | 80<br>110<br>150 | 55.5 |
| Mn0.1Ce0.5Zr0.4 | 60% Ce(Mn,Zr)$O_2$<br>40% t-Zr(Ce,Mn)$O_2$ | 5.371<br>5.213 | 100<br>70 | 87 |
| Mn0.2Ce0.5Zr0.3 | 46% Ce(Mn,Zr)$O_2$<br>52% t-Zr(Ce,Mn)$O_2$<br>2% Mn$_2$O$_3$ | 5.391<br>5.297<br>- | 70<br>40<br>230 | 103 |

* for t-Zr$O_2$ lattice parameter was calculated as a * = (2 × a × a × c)$^{1/3}$.

### 2.2. Variation of Zr/(Ce + Zr) Molar Ratio

The influence of the Zr/(Ce + Zr) molar ratio on the catalytic properties was studied for the series of Mn0.3(Zr,Ce)0.7 catalysts (Zr/(Ce + Zr) molar ratio was in the range from 0 to 1 at constant Mn content of 0.3). Figure 3a,c show the temperature dependence of CO and C$_3$H$_8$ conversion a for Mn0.3(Zr,Ce)0.7 catalysts. Figure 3b,d illustrate the temperature of 50% conversion of CO and C$_3$H$_8$ depending on the Zr/(Ce + Zr) molar ratio. For ternary Mn0.3(Zr,Ce)0.7 catalysts, similar catalytic properties in a wide range of cerium and zirconium contents are observed. Mn0.3(Zr,Ce)0.7 catalysts have a slightly higher catalytic activity in the CO oxidation reaction compared to the double oxides Mn0.3Zr0.7 and Mn0.3Ce0.7.

The value of T$_{50}$ is in the range of 159–164 °C for catalysts with molar ratio Zr/(Zr + Ce) = 0.14–0.71 and 178 and 172 °C for Mn0.3Ce0.7 and Mn0.3Zr0.7, respectively. Catalytic tests of the studied series of samples in the propane oxidation reaction showed similar results. Mn0.3(Zr,Ce)0.7 catalysts with the molar ratio Zr/(Zr + Ce) of 0.14–0.71 are more active than double oxides Mn0.3Zr0.7 and Mn0.3Ce0.7. The value of T$_{50}$ is in the range of 258–284 °C for catalysts with molar ratio Zr/(Zr + Ce) = 0.14–0.71 and 309 and 314 °C for Mn0.3Ce0.7 and Mn0.3Zr0.7, respectively. The Mn0.3Zr0.6Ce0.1 catalyst is characterized by the lowest activity in the propane oxidation reaction and similar activity to Mn0.3Ce0.7 and Mn0.3Zr0.7 in the CO oxidation reaction; T$_{50}$ of this sample is 355 °C and 176 °C, respectively. The reasons for this behavior will be discussed further.

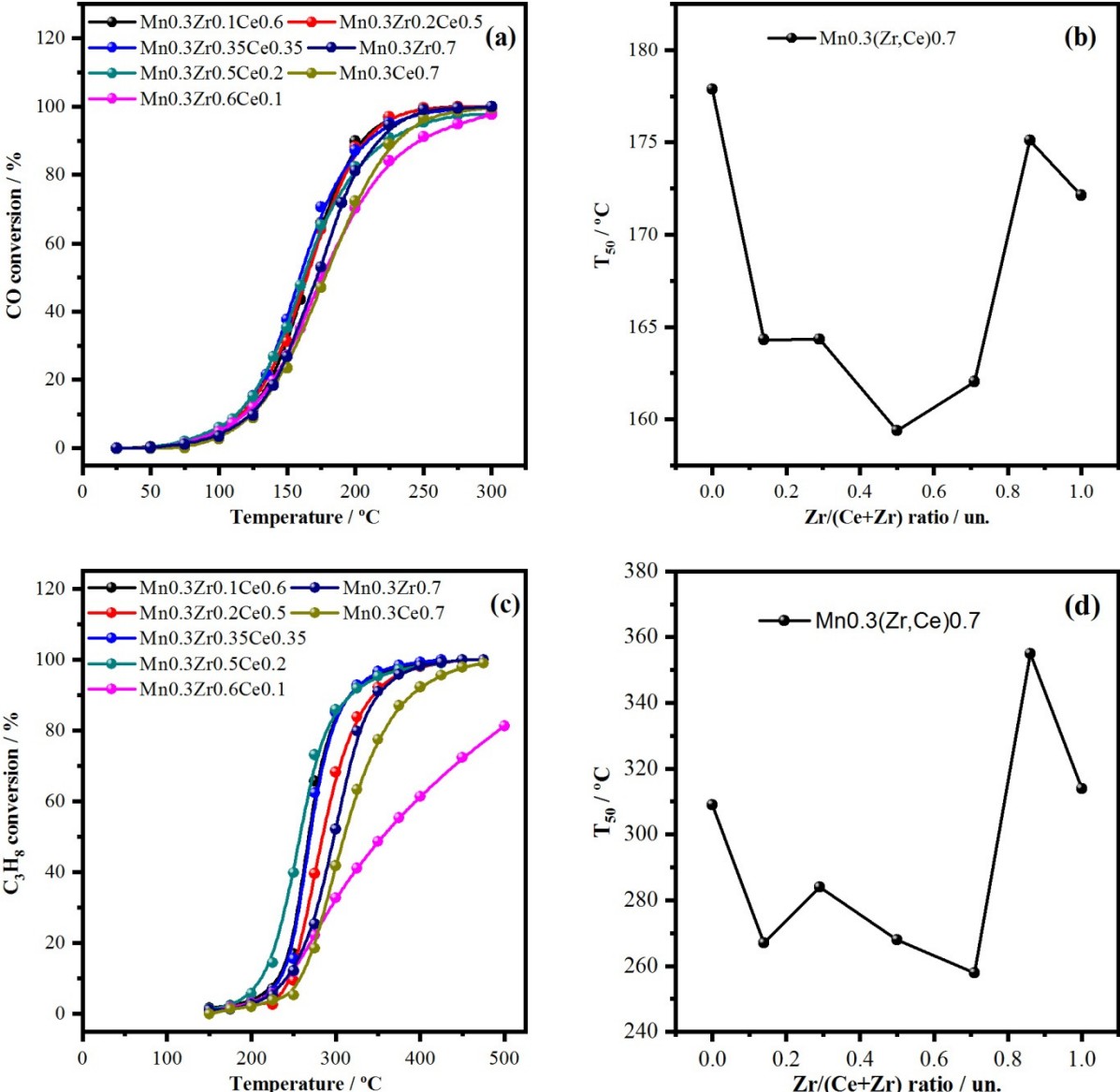

**Figure 3.** Reaction temperature dependence of (**a**) CO and (**c**) $C_3H_8$ conversion; the molar ratio $Zr/(Ce + Zr)$ dependence of $T_{50}$ for (**b**) CO and (**d**) $C_3H_8$ conversion in oxidation reaction over Mn0.3(Zr,Ce)0.7 catalysts.

XRD patterns of Mn0.3(Zr,Ce)0.7 catalysts are shown in Figure 4. Similarly to the (Mn,Zr)0.5Ce0.5 catalysts series, phases based on cubic $CeO_2$ and tetragonal $ZrO_2$ are observed. All the catalysts also contained some amount of $Mn_2O_3$. The cubic lattice parameter gradually shifts from 5.383 to 5.100 for Mn0.3Zr0.1Ce0.6 and Mn0.3Zr0.7, respectively (Table 2). Moreover, an increase in the zirconium content leads to a partial amorphization of phases. The most pronounced effect is for the Mn0.3Zr0.6Ce0.1 catalyst, which could be a reason for its low catalytic activity. In general, the Mn0.3(Zr,Ce)0.7 catalysts have a higher catalytic activity in the CO and propane oxidation reaction compared to the double oxide catalyst, such as Mn0.3Zr0.7 and Mn0.3Ce0.7. It is likely that the increased activity of ternary systems compared to binary systems is associated with additional restructuration of the solid solution (formation of ternary $Zr(Ce,Mn)O_2$ oxides, a decrease in crystallite size, appearance of second phase).

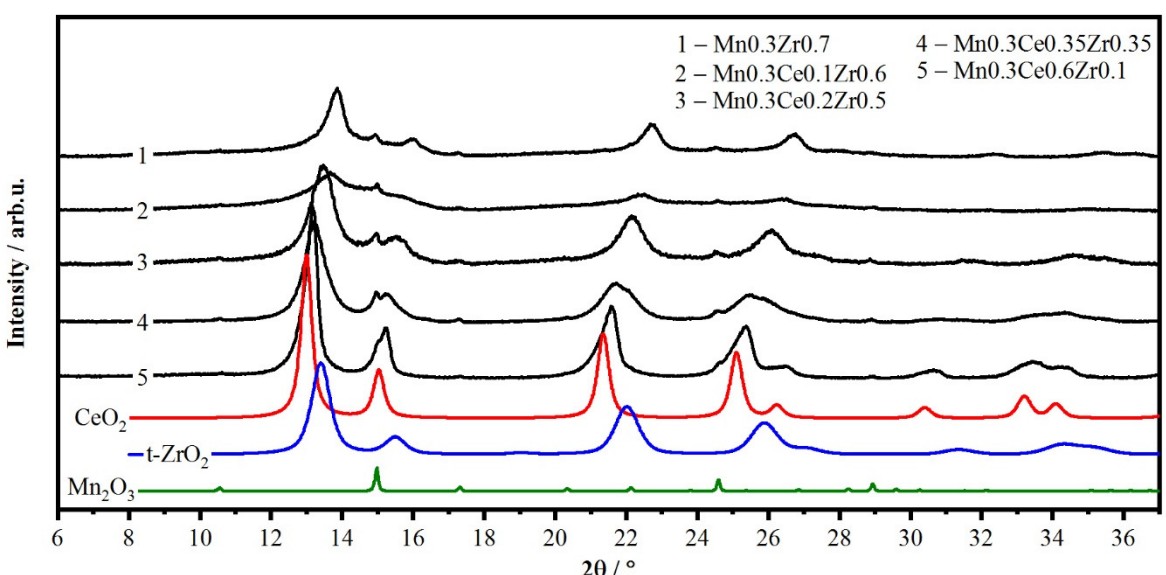

**Figure 4.** XRD patterns of Mn0.3(Zr,Ce)0.7 catalysts and peak positions of phases $CeO_2$ (PDF No. 431002), t-$ZrO_2$ (PDF No. 42-1164), and $Mn_2O_3$ (PDF No. 41-1442).

**Table 2.** Phase composition, lattice parameter of solid solution, CSR, and surface area of Mn0.3(Zr,Ce)0.7 catalysts.

| Catalyst | Phase Content, % Wt. | Lattice Parameter *, Å | CSR, Å | $S_{BET}$, m²/g |
|---|---|---|---|---|
| Mn0.3Ce0.6Zr0.1 | 20% amorphous $ZrO_2$<br>74% Ce(Mn,Zr)$O_2$<br>6% $Mn_2O_3$ | -<br>5.383<br>- | -<br>90<br>110 | 57.5 |
| Mn0.3Ce0.5Zr0.2 | 53% Ce(Mn,Zr)$O_2$<br>43% t-Zr(Ce,Mn)$O_2$<br>4% $Mn_2O_3$ | 5.383<br>5.307<br>- | 80<br>30<br>270 | 91 |
| Mn0.3Ce0.35Zr0.35 | 66% Ce(Mn,Zr)$O_2$<br>29% c-Zr(Ce,Mn)$O_2$<br>5% $Mn_2O_3$ | 5.338<br>5.162<br>- | 70<br>50<br>310 | 132 |
| Mn0.3Ce0.2Zr0.5 | Amorphous<br>t-Zr(Ce,Mn)$O_2$<br>$Mn_2O_3$ | -<br>5.234<br>- | -<br>40<br>130 | 162 |
| Mn0.3Ce0.1Zr0.6 | Amorphous<br>$CeO_2$<br>$Mn_2O_3$ | -<br>-<br>- | -<br>-<br>- | 223 |
| Mn0.3Zr0.7 | 54% amorphous<br>40% t-Zr(Mn)$O_2$<br>6% $Mn_2O_3$ | -<br>5.100<br>- | -<br>80<br>90 | 176 |

* for t-$ZrO_2$ lattice parameter was calculated as a * = $(2 \times a \times a \times c)^{1/3}$.

### 2.3. Variation of Mn/(Mn + Ce + Zr) Ratio

The influence of the Mn/(Mn + Ce + Zr) molar ratio on the catalytic properties was studied for the series of Mn(x)Zr(0.5 − x/2)Ce(0.5 − x/2) catalysts (Mn/(Mn + Ce + Zr) molar ratio was in the range from 0.1 to 0.8 at constant Zr/Ce molar ratio of 1). Figure 5a,c show the temperature dependence of CO and $C_3H_8$ conversion a for Mn(x)Zr(0.5 − x/2)Ce(0.5 − x/2) catalysts. Figure 5b,d illustrate the temperature of 50% conversion of CO and $C_3H_8$ depending on the Mn/(Mn + Ce + Zr) molar ratio.

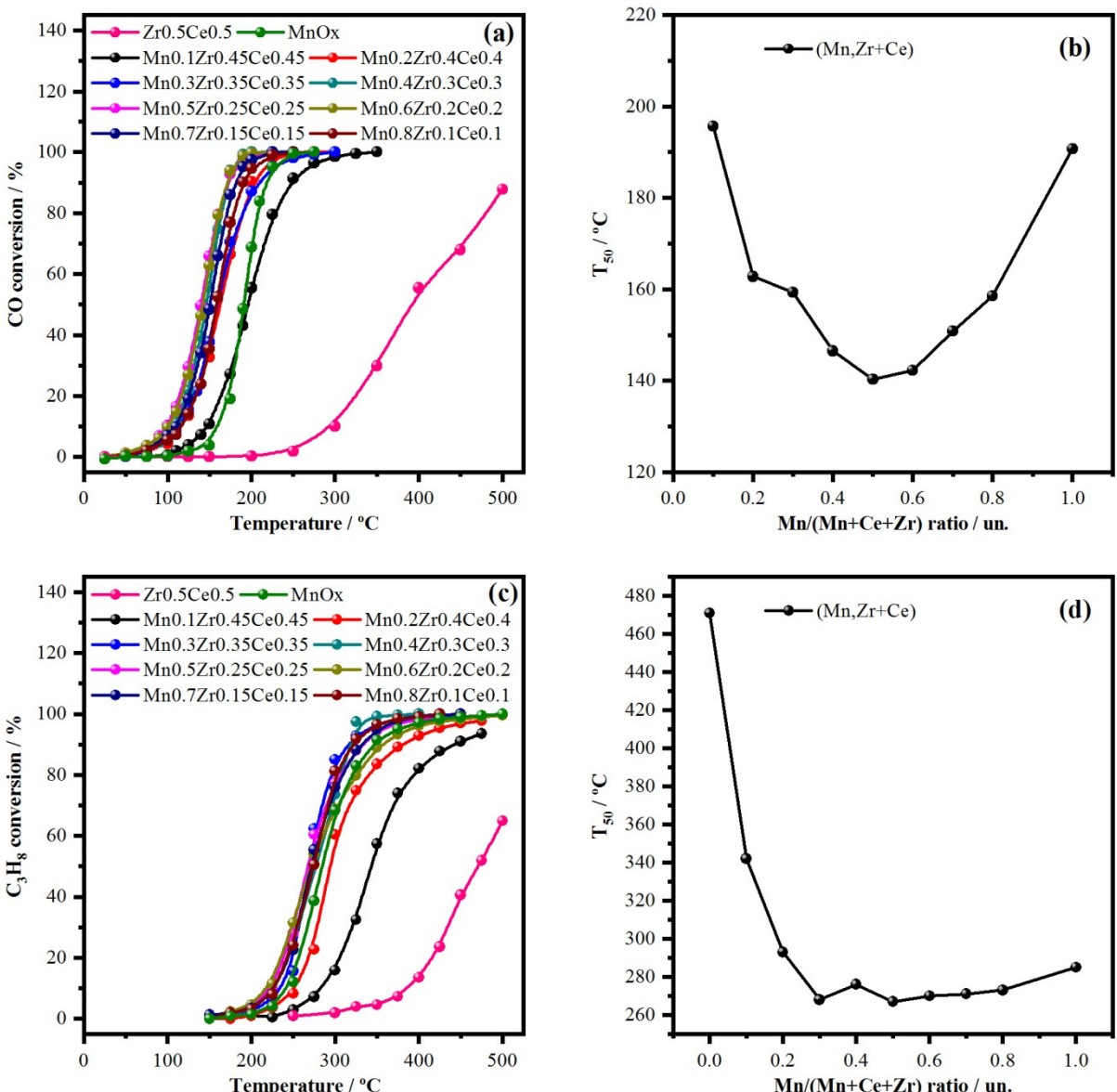

**Figure 5.** Reaction temperature dependence of (**a**) CO and (**c**) $C_3H_8$ conversion; the molar ratio Mn/(Mn + Ce + Zr) dependence of $T_{50}$ for (**b**) CO and (**d**) $C_3H_8$ conversion in oxidation reaction over Mn(x)Zr(0.5 − x/2)Ce(0.5 − x/2) catalysts.

Catalyst tests show an increase in catalytic activity with increasing Mn content from 0.1 to 0.5, as evidenced by the shift of the CO conversion curve to the low-temperature region. The dependence of $T_{50}$ on the manganese content in the CO oxidation reaction has a U-shape with a minimum for the Mn0.5Zr0.25Ce0.25 catalyst; $T_{50}$ varies in the range of 195 to 140 °C (Figure 5b). For the reference sample of pure manganese oxide, $T_{50}$ is 191 °C. For the reference sample Zr0.5Ce0.5, $T_{50}$ is more than 375 °C.

In the reaction of propane oxidation, similarly to the (Mn,Zr)0.5Ce0.5 catalysts series the introduction of manganese into the Zr-Ce system is accompanied by a significant increase in activity with a plateau for catalysts with Mn/(Mn + Zr + Ce) > 0.3. The increase in the Mn/(Mn + Ce + Zr) ratio up to 0.3 leads to a rapid decrease in the $T_{50}$ from 471 to 268 °C (Figure 5d). For other compositions, $T_{50}$ fluctuates in the range of 267–276 °C. For the reference sample $MnO_x$, a slight decrease in the degree of propane conversion is observed ($T_{50}$ = 285 °C). For the reference sample Zr0.5Ce0.5, $T_{50}$ is more than 450 °C.

Therefore, the catalytic activity ($T_{50}$) behavior for the Mn(x)Zr(0.5 − x/2)Ce(0.5 − x/2) catalysts in the propane and in the CO oxidation reaction differs significantly. It is

worth mentioning that activity in propane oxidation of pure manganese oxide is close to those for Mn-Ce-Zr catalysts with Mn/(Mn + Ce + Zr) = 0.3–0.8. Apparently, this is due to the lower sensitivity of the propane oxidation reaction to the environment of manganese than to its amount.

The crystal phases of Mn(x)Zr(0.5 − x/2)Ce(0.5 − x/2) catalysts were determined by XRD, and the results are shown in Figure 6 and Table 3. According to XRD data, with an increase in the Mn content in the catalyst, the content of crystalline manganese oxides such as $Mn_2O_3$ and $Mn_3O_4$ increases and the amount of the solid solution decreases. For Mn0.3Ce0.35Zr0.35, the content of manganese oxide was 6 wt%, while for Mn0.8Ce0.1Zr0.1 it reaches 67 wt%. Simultaneously, the concentration of manganese and, accordingly, oxygen vacancies in the composition of mixed oxides increases (details of estimation of solid solution composition is given in Supplementary Materials).

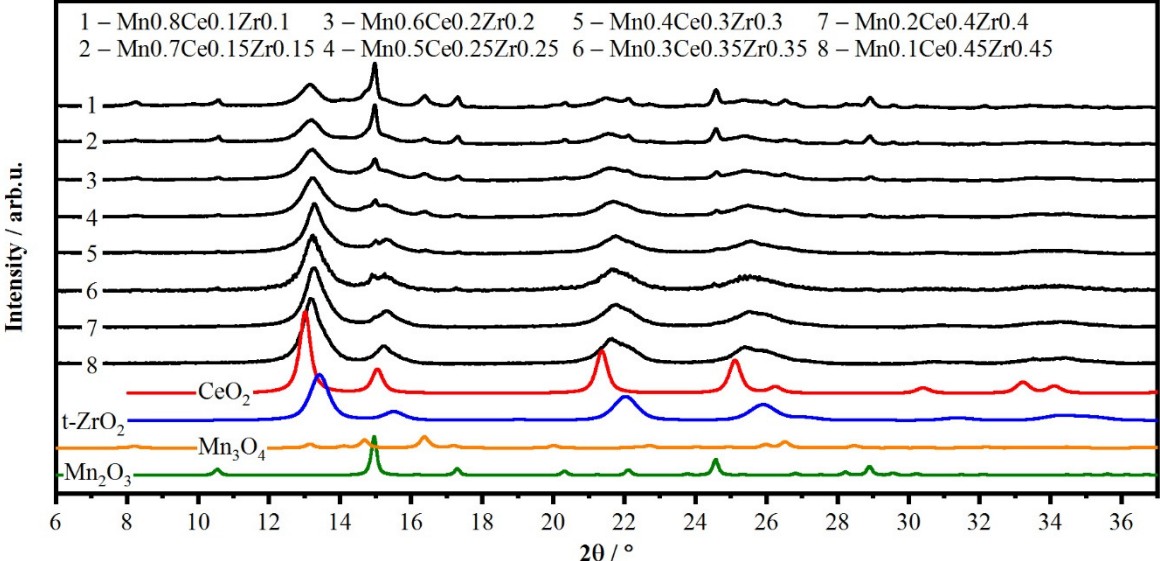

**Figure 6.** XRD patterns of Mn(x)Zr(0.5 − x/2)Ce(0.5 − x/2) catalysts and peak positions of phases $CeO_2$ (PDF No. 431002), t-$ZrO_2$ (PDF No. 42-1164), $Mn_2O_3$ (PDF No. 41-1442), and $Mn_3O_4$ (PDF No. 24-0734).

Figure 7 shows TPR-$H_2$ curves and total amount of hydrogen consumed for the catalysts Mn(x)Zr(0.5 − x/2)Ce(0.5 − x/2) by varying the Mn/(Ce + Zr + Mn) ratio from 0.1 to 0.6. For samples with a low manganese content (Mn0.1Zr0.45Ce0.45 and Mn0.2Zr0.4Ce0.4), a wide halo was observed in the temperature range of 160–520 °C. In this range, the TPR profiles contain a broad signal of partially overlapped peaks with the main maxima at ~260–270 °C and 420 °C. These signals are associated with the consumption of hydrogen due to the stepwise partial reduction of manganese cations present in the Zr-Ce-containing solid solution [23–25]. The signal at ~260–270 °C corresponds to the reduction of $Mn^{4+}/Mn^{3+}$ ions to $Mn^{3+}/Mn^{2+}$, while the peak at ~350–360 °C shows $Mn^{3+}/Mn^{2+}$ → $Mn^{2+}$ transition [26–28]. On the other hand, the consumption of hydrogen in the region at 150–300 °C can be explained by the reduction of finely dispersed $MnO_x$ particles, which can be present on the surface of the solid solution [23,24]. As Mn content increases, more pronounced peaks with maxima at 304 and 388 °C appear, indicating the two-stage reduction of $Mn_2O_3$ to $Mn_3O_4$ and then to MnO [28], which is present according XRD data. There is also an almost linear increase in the total amount of hydrogen consumption with an increase in the manganese content (Figure 7b). The value grows from 1.44 to 3.51 mmol ($H_2$)/g. It is worth mentioning that the amount of hydrogen consumption before 300 °C gradually grows from 0.35 to 1.37 mmol ($H_2$)/g with an increase in the Mn/(Ce + Zr + Mn) ratio from 0.1 to 0.5. Further increase in the Mn/(Ce + Zr + Mn) ratio to 0.6 leads to a decrease of the amount of adsorbed $H_2$ before 300 °C to the value of 1.21 mmol ($H_2$)/g. The

amount of low-temperature hydrogen consumption correlates well with catalytic activity (Figures 5 and S1).

**Table 3.** Phase composition, lattice parameter of solid solution, CSR, and surface area for Mn(x)Zr(0.5 − x/2)Ce(0.5 − x/2)catalysts.

| Catalyst | Phase Content, % Wt. | Lattice Parameter *, Å | CSR, Å | $S_{BET}$, m²/g |
|---|---|---|---|---|
| Mn0.1Ce0.45Zr0.45 | 47% Ce(Mn,Zr)O₂<br>53% t-Zr(Ce,Mn)O₂ | 5.351<br>5.216 | 70<br>50 | 106 |
| Mn0.2Ce0.4Zr0.4 | 64% Ce(Mn,Zr)O₂<br>36% t-Zr(Ce,Mn)O₂ | 5.328<br>5.210 | 60<br>50 | 123 |
| Mn0.3Ce0.35Zr0.35 | 47% Ce(Mn,Zr)O₂<br>47% t-Zr(Ce,Mn)O₂<br>6% Mn₂O₃ | 5.352<br>5.220<br>- | 60<br>40<br>170 | 132 |
| Mn0.4Ce0.3Zr0.3 | 37% Ce(Mn,Zr)O₂<br>55% t-Zr(Ce,Mn)O₂<br>4% Mn₂O₃<br>4% Mn₃O₄ | 5.314<br>5.258<br>-<br>- | 75<br>20<br>220<br>170 | 138.5 |
| Mn0.5Ce0.25Zr0.25 | 34% Ce(Mn,Zr)O₂<br>48% t-Zr(Ce,Mn)O₂<br>7% Mn₂O₃<br>11% Mn₃O₄ | 5.330<br>5.234<br>-<br>- | 60<br>20<br>225<br>140 | 141.3 |
| Mn0.6Ce0.2Zr0.2 | 29% Ce(Mn,Zr)O₂<br>43% Zr(Ce,Mn)O₂<br>12% Mn₂O₃<br>15% Mn₃O₄ | 5.345<br>5.189<br>-<br>- | 60<br>20<br>220<br>125 | 150.3 |
| Mn0.7Ce0.15Zr0.15 | 36% Ce(Mn,Zr)O₂<br>14% Zr(Ce,Mn)O₂<br>38% Mn₂O₃<br>12% Mn₃O₄ | 5.364<br>4.922<br>-<br>- | 50<br>230<br>160<br>40 | 118 |
| Mn0.8Ce0.1Zr0.1 | 22% Ce(Mn,Zr)O₂<br>11% Zr(Ce,Mn)O₂<br>30% Mn₂O₃<br>22% Mn₃O₄<br>15% MnO₂ | 5.371<br>4.961<br>-<br>-<br>- | 50<br>50<br>245<br>160<br>55 | 92.7 |
| MnOₓ | Mn₂O₃ | - | 210 | |

* for t-ZrO₂, lattice parameter was calculated as a * = (2 × a × a × c)$^{1/3}$.

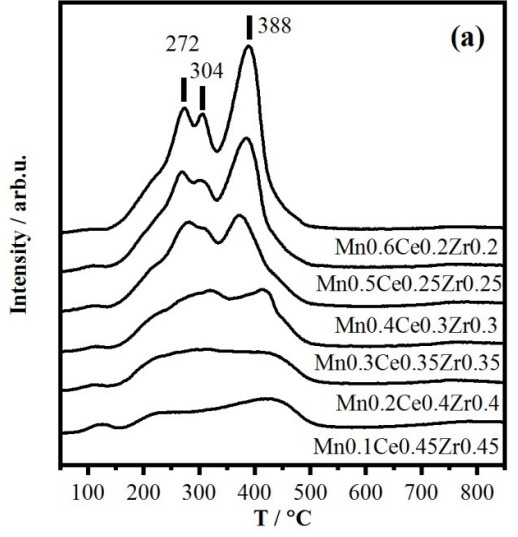

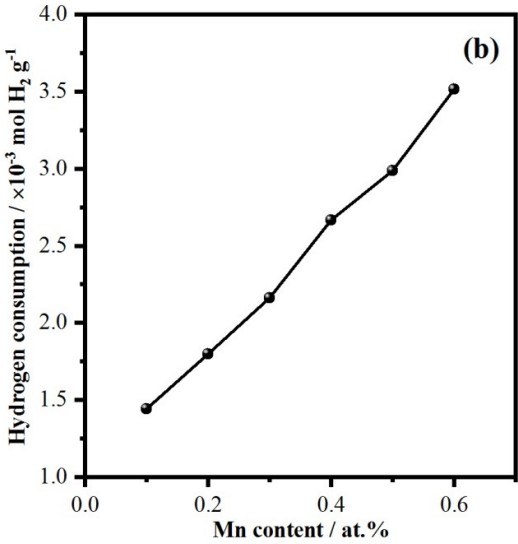

**Figure 7.** TPR curves (**a**) and total amount of hydrogen consumed per g (**b**) of Mn(x)Zr(0.5 − x/2) Ce(0.5 − x/2) catalysts.

The XPS was used to evaluate changes in the electronic properties and ratios between the elements on the surface of the catalysts. Table 4 summarizes the relative concentrations (atomic ratios) of the elements in the surface of the catalysts. Figure 8 shows the Zr3d, Ce3d, and Mn2p XPS core-level spectra of Mn(x)Zr(0.5 − x/2)Ce(0.5 − x/2) catalysts. The Zr3d spectra are fitted by $Zr3d_{5/2}$-$Zr3d_{3/2}$ doublet with the $Zr3d_{5/2}$ binding energy of 181.9 eV. The binding energy of $Zr3d_{5/2}$ peak corresponds to zirconium in the $Zr^{4+}$ state [29–31]. The Ce3d spectra of the catalysts have a complex structure of peaks corresponding to $Ce^{3+}$ and $Ce^{4+}$ states [32,33]. The fitting into individual components allows one to evaluate the fraction of $Ce^{3+}$ cations (Table 4). It was found that cerium predominantly exists in the form of $Ce^{4+}$, while the fraction of $Ce^{3+}$ ions vary in the range of 16–22%.

**Table 4.** Atomic ratios of the elements on the surface according to XPS results.

| Catalyst | [Mn]/[Mn + Ce + Zr] | | | | [O]/[Me] | $Ce^{3+}$, % |
|---|---|---|---|---|---|---|
| | Total | %, $Mn^{2+}$ | %, $Mn^{3+}$ | %, $Mn^{4+}$ | | |
| Mn0.1Ce0.45Zr0.45 | 0.13 | 92 | 8 | 0 | 2.31 | 22 |
| Mn0.2Ce0.4Zr0.4 | 0.27 | 51 | 49 | 0 | 2.35 | 20 |
| Mn0.3Ce0.35Zr0.35 | 0.49 | 22 | 77 | 1 | 1.86 | 18 |
| Mn0.4Ce0.3Zr0.3 | 0.51 | 29 | 64 | 7 | 1.98 | 18 |
| Mn0.5Ce0.25Zr0.25 | 0.58 | 14 | 80 | 6 | 1.80 | 16 |
| Mn0.6Ce0.2Zr0.2 | 0.65 | 20 | 70 | 10 | 1.76 | 20 |

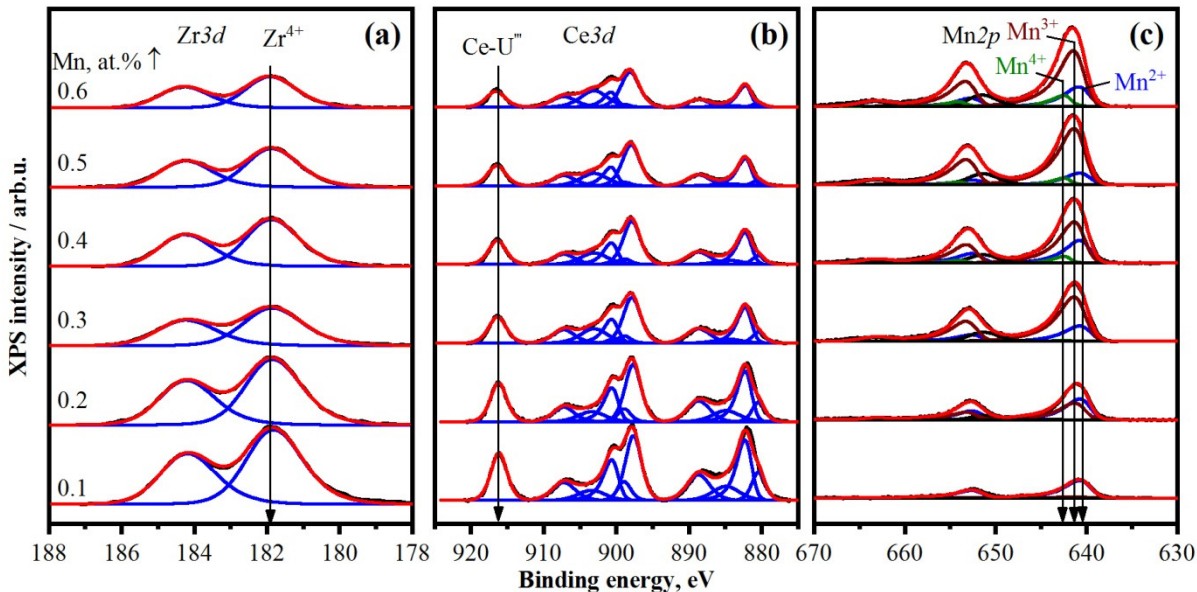

**Figure 8.** XPS Zr3d (**a**), Ce3d (**b**), and Mn2p (**c**) core-level spectra of the Mn(x)Zr(0.5 − x/2)Ce(0.5 − x/2) catalysts. Ce3d and Mn2p spectra are normalized to the integral intensity of the corresponding Zr3d spectra.

To identify the chemical state of manganese, the binding energy of the $Mn2p_{3/2}$ peak and the intensity and position of satellites are used [34–36]. The asymmetry of main peaks is caused by the multielectron process. The fitting of the Mn2p spectra allows us to evaluate the fraction of manganese cations in the different oxidation states: $Mn^{2+}$, $Mn^{3+}$, and $Mn^{4+}$. The Mn2p spectra are fitted by three $Mn2p_{3/2}$–$Mn2p_{1/2}$ doublets corresponding to manganese in the $Mn^{2+}$, $Mn^{3+}$, and $Mn^{4+}$ states, and by the satellites located, respectively, at a distance of 6.7, 10.6, and 11.8 eV from the main peaks. The binding energies of $Mn2p_{3/2}$ peaks equal to 639.0, 640.7, and 641.4 eV correspond to the $Mn^{2+}$, $Mn^{3+}$, and $Mn^{4+}$ states, respectively. The $Mn2p_{3/2}$ binding energies reported for MnO, $Mn_2O_3$, and $MnO_2$ oxides are in the ranges of 640.4–641.7, 641.5–641.9, and 642.2–642.6 eV, respectively [34,35,37–39].

The XPS analysis allows us to estimate the fraction of manganese in different oxidation states (Table 4). The relative surface concentrations (atomic ratios) of elements are also listed in Table 4,. With an increase in the Mn/(Mn + Ce + Zr) ratio from 0.1 to 0.6, the surface [Mn]/[Mn + Ce + Zr] ratio linearly increases from 0.13 to 0.65. Simultaneously, the average oxidation state of Mn grows from 2.1 to 2.9. With an increase in the surface concentration of Mn, a redistribution of the charge state of cations from 2+ to 4+ is observed.

## 3. Discussion

The ternary oxides Mn0.3(Zr,Ce)0.7 demonstrate similar catalytic properties in a wide range of cerium and zirconium contents without a clearly pronounced dependence. In general, Mn0.3(Zr,Ce)0.7 catalysts have a higher catalytic activity in the CO and propane oxidation reaction compared to the binary oxides Mn0.3Zr0.7 and Mn0.3Ce0.7. As the manganese content increases and the zirconium content decreases accordingly, the catalytic activity of (Mn,Zr)0.5Ce0.5 ternary oxides gradually increases. The highest catalytic activity in the CO oxidation reaction is achieved for the composition Mn0.4Zr0.1Ce0.5. Mn(x)Zr(0.5 − x/2)Ce(0.5 − x/2) shows different behavior in the reaction of CO and propane oxidation. The dependence of the catalytic activity in CO oxidation for Mn(x)Zr(0.5 − x/2)Ce(0.5 − x/2) on the manganese content has a pronounced extremum, the highest activity is observed for the composition Mn0.5Zr0.25Ce0.25. In the propane oxidation reaction, the activity of a series of catalysts Mn(x)Zr(0.5 − x/2)Ce(0.5 − x/2) increases with an increase in manganese concentration to 0.3, and with further introduction of manganese remains at a constant level. Catalytic activity in propane oxidation is comparable for pure $Mn_2O_3$ and ternary Mn(x)Zr(0.5 − x/2)Ce(0.5 − x/2) catalysts (Figure 5d). Differences in the properties of Mn(x)Zr(0.5 − x/2)Ce(0.5 − x/2) samples with varying manganese content in CO and propane oxidation reactions can be associated with different mechanisms of these two processes. The reaction of CO oxidation to $CO_2$ is a simpler process, the activity of which is primarily determined by the ability of the catalyst to activate oxygen [40]. In this case, the catalytic activity is determined by the content of manganese in the composition of the Mn-Zr-Ce solid solution, since the incorporation of manganese cations into the lattice of cerium and zirconium oxides is accompanied by the formation of lattice defects and oxygen vacancies, which increases the ability of the compound to accumulate oxygen and increases the oxygen mobility on the catalyst surface [41]. The propane oxidation reaction is a more complex process. Generally, propane oxidation on Mn-based catalysts follows two major steps: dehydrogenation of propane and deep oxidation of oxygenated intermediate species (aldehydes, carboxylic acid, carbonate) to $CO_2$ and $H_2O$ [42]. Therefore, the catalyst must be able to activate C–H bonds and promote deep oxidation. C–H cracking is the rate-determining step for propane oxidation, so propane adsorption on the catalyst surface and its activation play a crucial role in achieving high activity [43]. In addition, according to Hu et al. [43], weakly bound oxygen and chemically adsorbed oxygen play a key role in the propane oxidation reaction, in contrast to the process of carbon monoxide oxidation.

When considering the surface area that determines quantity of active sites, one would expect that the activity will be governed by the surface area of the catalysts. Therefore, samples with the highest surface area should be the most active ones. However, although the Mn0.3Ce0.1Zr0.6 catalyst possesses the highest surface area of 223 $m^2$/g (Table 2), its activity is not the best ($T_{50}$ ($C_3H_8$) is 268 °C, $T_{50}$ (CO) is 175 °C). Figures S2 and S3 show that there is no dependence between surface area and $T_{50}$ ($C_3H_8$) and $T_{50}$ (CO). Since our catalyst consists of different phases, we also tried to find a correlation between a certain phase and catalytic activity. Figures S4 and S5 show $T_{50}$ in CO and propane oxidation reactions versus content of $Mn_xZr_yCe_{1-x-y}O_2$ solid solution and crystalline manganese oxides. In the case of CO, there are no clear dependencies. When the content of solid solutions is 90–100%, both the largest and smallest values of $T_{50}$ are observed (for Mn0.4Ce0.3Zr0.3 and Mn0.1Ce0.45Zr0.45 catalyst $T_{50}$ is 147 and 196 °C, respectively). However, in the case of propane oxidation, an increase in the content of Mn oxide to 15–20 wt% leads to an increase in the catalytic activity. With further growth of Mn oxide content, $T_{50}$ changes in

the narrow range of temperatures from 267 to 290 °C. The content of manganese (x) in the composition of the $Mn_xZr_yCe_{1-x-y}O_2$ solid solution was also estimated (the procedure of x estimation is given in Supporting Information). Figures S6 and S7 show the correlation between activity and x in $Mn_xZr_yCe_{1-x-y}O_2$ solid solution. In the case of CO oxidation, with an increase in the x content in $Mn_xZr_yCe_{1-x-y}O_2$ solid solution, a decrease in $T_{50}$ is observed. As x increases, the concentration of oxygen vacancies is expected to increase accordingly. According to XPS analysis, catalysts contain $Mn^{2+}$, $Mn^{3+}$, $Mn^{4+}$, $Ce^{3+}$, $Ce^{4+}$, and $Zr^{4+}$, and the average oxidation state of Mn is 2.1–2.9 (Table 4). The presence of cations in oxidation state less than 4+ implies the creation of an oxygen vacancy, based on the electroneutrality principle, which, in turn, enhances the adsorption capacity of reactive oxygen species on the catalyst surface [44]. According to TPR data, low-temperature reducibility of catalyst (before 300 °C) correlates well with catalytic activity (Figure S1). The maximum activity in CO oxidation likely corresponds to the balance between the fraction of the solid solution and the concentration of manganese in it. Crystalline manganese oxides seem to make a smaller contribution to the activity of the catalyst in the reaction of CO oxidation. At the same time, the propane oxidation reaction is less sensitive to the environment of manganese cations than to its amount. A decrease in the content of the solid solution is compensated for by an increase in the content of crystalline phases of manganese oxides and an increase in the surface content of manganese, thus maintaining high activity values. It should be noted that the catalysts obtained in this work are comparable to catalytic activity in CO and propane oxidation reactions with oxide ternary Mn-Zr-Ce systems described by other researchers (Table 5).

**Table 5.** Conversion of propane and CO on oxide ternary Mn-Zr-Ce catalysts reported in recent publications.

| Catalyst | Preparation Method | Calcination Temperature, °C | Concentration of $C_3H_8$ (CO) | GHSV, mL/(g*h) | $T_{50}$, °C | Reference |
|---|---|---|---|---|---|---|
| | | | Propane Oxidation | | | |
| $Ce_{0.76}Zr_{0.19}Mn_{0.05}O_{(2-x)}$ | co-precipitation | 650 | 1 mol.% | 50,000 | 408 | [6] |
| $Mn_{0.3}Ce_{0.2}Zr_{0.5}$ | co-precipitation | 600 | 1 vol.% | 37,200 | 258 | This work |
| | | | CO Oxidation | | | |
| $Mn_{0.45}Ce_{0.45}Zr_{0.1}$ | co-precipitation | 550 | 4 vol.% | 1800 | 105 | [45] |
| $MnO_x–Ce_{0.8}Zr_{0.2}O_2$ | co-precipitation | 500 | 2 vol.% | 36,000 | 160 | [12] |
| $Mn_{0.5}Ce_{0.25}Zr_{0.25}$ | co-precipitation | 600 | 1 vol.% | 58,440 | 140 | This work |

## 4. Experimental

### 4.1. Catalyst Preparation

The samples were prepared by the co-precipitation method. A joint solution of $ZrO(NO_3)_2$, $Ce(NO_3)_3$, and $Mn(NO_3)_2$ salts was obtained, and the $NH_4OH$ solution was gradually added into the mixture with continual stirring until the pH reached 10. The precipitation process was carried out at 80 °C. Stirring of the suspension was continued for 1 h, then $H_2O_2$ was added dropwise to ensure the completeness of precipitation. The amount of $H_2O_2$ corresponded to the molar ratio of $H_2O_2$:(Mn + Zr + Ce) = 1. The suspension was finally kept without stirring for 2 h. The obtained precipitate was filtered off and washed with distilled water on the filter to pH = 6–7. The samples were dried at 120 °C for 2 h and calcined in a muffle furnace at 600 °C for 4 h. Three series of Mn-Ce-Zr catalysts were synthesized by co-precipitation with varying (1) Zr/Zr + Ce molar ratio at constant manganese content of 0.3; (2) manganese content at constant Zr/Ce molar ratio of 1; (3) Mn/Mn + Zr molar ratio at constant Ce content of 0.5.

The samples are designated as MnxZryCez, where x, y, z are the molar fraction of Mn, Zr, Ce, and x + y + z = 1.

### 4.2. Catalyst Characterization

The phase composition was determined by XRD using a STOE STADI MP diffractometer equipped with a Mythen2 1K (Dectris, Baden, Switzerland) linear detector. The diffraction patterns were obtained in transmission $\theta/2\theta$ geometry in the $2\theta$ range from 3 to $50°$ with a step of $0.015°$ using monochromatic $MoK_{\alpha 1}$ radiation ($\lambda = 0.7093$ Å, Ge crystal monochromator). Rietveld refinement for quantitative analysis was carried out using the software package Topas V.4.2.

The specific surface area was calculated with the Brunauer–Emmett–Teller (BET) method using nitrogen adsorption isotherms measured at liquid nitrogen temperatures on an automatic Micromeritics ASAP 2400 sorptometer (Norcross, GA, USA).

The temperature-programmed reduction in hydrogen (TPR-$H_2$) was performed with 40–60 mg of sample in a quartz reactor using a flow setup with a thermal conductivity detector. The reducing mixture (10 vol.% of $H_2$ in Ar) flow was 40 mL/min. The heating rate from room temperature to $900\ °C$ was $10\ °C$/min. The TPR curves were normalized per catalyst mass.

The XPS measurements were performed on a photoelectron spectrometer (SPECS Surface Nano Analysis GmbH, Berlin, Germany) equipped with a PHOIBOS-150 hemispherical electron energy analyzer and an XR-50 X-ray source with a double Al/Mg anode. The core-level spectra were obtained using $AlK\alpha$ radiation ($h\nu = 1486.6$ eV) under ultrahigh vacuum conditions. The binding energy ($E_b$) of photoemission peaks was corrected for the $Ce3d_{3/2}$-U''' peak ($E_b = 916.7$ eV) of cerium oxide. The curve fitting was done by the CasaXPS software [46]. The line shape used in the fit was the sum of Lorentzian and Gaussian functions. A Shirley-type background was subtracted from each spectrum [47]. Relative element concentrations were determined from the integral intensities of the core-level spectra using the theoretical photoionization cross-sections according to Scofield [48].

### 4.3. Catalyst Tests

Catalyst samples were tested in CO and propane oxidation reactions in a glass reactor (170 mm $\times$ Ø 10 mm) at atmospheric pressure in a flow mode. Following that, 0.5 g of the catalyst (fraction 0.4–0.8 mm) was diluted with an inert $SiO_2$ sand (up to 3.0 mL). The temperature in the catalyst bed was controlled and regulated using a chromel-alumel thermocouple connected to a heat controller.

In the case of CO oxidation reaction, the gas mixture contained 1 vol.% of CO and 99 vol.% of air. The total flow rate was 487 mL/min. Catalytic tests in the CO oxidation reaction were carried out in the temperature range from 30 to $500\ °C$. The analysis of the inlet and outlet reaction flows were carried out using a gas chromatograph. The mixture was separated on a packed column filled with CaA zeolite (3 m). The unreacted amount of CO was determined using a thermal conductivity detector. CO conversion ($X_{CO}$) was calculated by the equation:

$$X_{CO} = |(P_{CO}/P_{N2})^{in} - (P_{CO}/P_{N2})^{out}|/(P_{CO}/P_{N2})^{in} \tag{1}$$

where $P_{CO}$ and $P_{N2}$ are the peak areas on the chromatogram corresponding to the CO and $N_2$ concentrations in the inlet ($^{in}$) and outlet ($^{out}$) flows. $P_{N2}$ was used as an internal standard.

Catalytic tests in the propane oxidation reaction were carried out in the temperature range from 150 to $500\ °C$. The reaction gas mixture contained 1 vol.% of $C_3H_8$ and 99 vol.% of air and was fed into the reactor at a rate of 310 mL/min. The inlet and outlet reaction flows were analyzed using a Tsvet-800 gas chromatograph supplied with a capillary column (filled with $SiO_2$) and a flame ionization detector. Nitrogen was used as the carrier gas. The conversion of propan ($X_{C3H8}$) was calculated from the chromatographic data using the following equation:

$$X_{C3H8} = (P^{in} - P^{out})/P^{in} \tag{2}$$

where $P^{in}$ and $P^{out}$ are the peak areas on the chromatogram corresponding to the propane concentration in the inlet ($^{in}$) and outlet ($^{out}$) flows.

## 5. Conclusions

The influence of the cation ratio in the $MnO_x$-$ZrO_2$-$CeO_2$ catalysts on their structural properties and catalytic activity in the CO and $C_3H_8$ oxidation reactions has been studied. It was shown that ternary Mn0.3(Zr,Ce)0.7 catalysts have a higher catalytic activity in the CO and propane oxidation reaction compared to the double oxide catalyst such as Mn0.3Zr0.7 and Mn0.3Ce0.7. The change in Zr/(Zr + Ce) mole ratio does not significantly affect the catalytic properties. The maximum activity in CO oxidation corresponds to the balance between the amount of the solid solution and the concentration of manganese in it. The best catalytic performance is exhibited by the catalyst with Mn/(Mn + Zr + Ce) = 0.5. In the case of propane oxidation reaction, there is «sigma» like dependence. Activity increases with manganese concentration to Mn/(Mn + Zr + Ce) = 0.3, and the activity remains constant with further introduction of manganese. The propane oxidation reaction is less sensitive to the environment of manganese cations than to its amount.

**Supplementary Materials:** The following supporting information can be downloaded at: https://www.mdpi.com/article/10.3390/catal13010211/s1, Figure S1: Temperature of 50% CO conversion (blue) and $H_2$ consumption before 300 °C (red) versus the Mn content; Figure S2: Temperature of 50% CO conversion versus the specific surface area ($S_{BET}$) for all catalysts; Figure S3: Temperature of 50% $C_3H_8$ conversion versus the specific surface area ($S_{BET}$) for all catalysts; Figure S4: Temperature of 50% CO conversion versus the content of solid solutions for all catalysts; Figure S5: Temperature of 50% $C_3H_8$ conversion versus the content of crystalline Mn oxides for all catalysts; Figure S6: Temperature of 50% CO conversion versus the content of Mn (x) in $Mn_xCe_yZr_{1-x-y}O_2$ solid solution for all catalysts; Figure S7: Temperature of 50% $C_3H_8$ conversion versus the content of Mn (x) in $Mn_xCe_yZr_{1-x-y}O_2$ solid solution for all catalysts; Table S1: List of catalysts; Table S2: Phase composition of catalysts.

**Author Contributions:** Conceptualization, O.A.B.; formal analysis, T.N.A., A.A.S., V.P.K., V.A.R., Z.S.V., D.V.Y. and E.E.A.; investigation, Z.S.V., T.N.A., A.A.S., V.P.K., V.A.R., D.V.Y. and E.E.A.; writing—original draft preparation, O.A.B. and Z.S.V.; writing—review and editing, O.A.B. and Z.S.V.; visualization, Z.S.V.; supervision, O.A.B. All authors have read and agreed to the published version of the manuscript.

**Funding:** This work was supported by the Russian Science Foundation, grant 21-73-10218.

**Acknowledgments:** The XRD and XPS studies were carried out using facilities of the shared research center "National Center of Investigation of Catalysts" at Boreskov Institute of Catalysis.

**Conflicts of Interest:** The authors declare no conflict of interest.

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
