# Peer review of "The Formation of Mn-Ce-Zr Oxide Catalysts for CO and Propane Oxidation: The Role of Element Content Ratio"

_catalysts, doi:10.3390/catal13010211_

Round 1
Reviewer 1 Report
The manuscpirt "The formation of Mn-Ce-Zr oxide catalysts for CO and propane oxidation: the role of cation content ratio" is devoted to design of mixed oixde catalysts for air purification from volatile organic polluants.
The presented results has high impact in understanding of active site formation in mixed oxide catalysts.
I have few questions and remarks:
1. Please check the writing of all formules (for example MnOx or MnOx?)
2. Experimental part: precipitationmethod of co-precipitation method was used?
3. XRD data: It can be seen from Fig. 1 that intensity of peaks of CeO2 is higher than one for ZrO2 phase. Why the loading of ZrO2 is higher than CeO2 for few sample (Table 1)?
4. Table 4. How the O/Me ration may be higher than 2 in mixed oxides?
5. Find the correlation between activity activity and structure for such mixed oxide catalysts is dificult. Probably the discussion of TPR data (for example in intensity of peaks at 200-300 oC) should be taken into account, because the main temperature of propane oxidation is in same region.
Author Response
Response to Review
We are thankful to the referee for useful comments. We made revision of the text according to the recommendations.
The manuscpirt "The formation of Mn-Ce-Zr oxide catalysts for CO and propane oxidation: the role of cation content ratio" is devoted to design of mixed oixde catalysts for air purification from volatile organic polluants.
The presented results has high impact in understanding of active site formation in mixed oxide catalysts.
I have few questions and remarks:
- Please check the writing of all formules (for example MnOx or MnOx?)
According to reviewer recommendation, the writings of all formulas has been checked. MnOx was chosen.
- Experimental part: precipitation method of co-precipitation method was used?
We used the co-precipitation method. The text has been corrected (page 2).
- XRD data: It can be seen from Fig. 1 that intensity of peaks of CeO2 is higher than one for ZrO2 phase. Why the loading of ZrO2 is higher than CeO2 for few sample (Table 1)?
Indeed, intensity of peaks of CeO2 is higher than one for ZrO2 phase (Figure 1), however results of Rietveld refinement shows opposite values (Table 1, for example, Ce0.5Zr0.5 sample). There two explanations of these phenomena: (1) different scattering factors of elements, (2) different widths of peaks. For example, in the case of Ce0.5Zr0.5 sample, decomposition of the first XRD peak by the PseudoVoigt function gives the following values of function parameters: peak area – 2836, peak maximum – 13.011, height – 4778, FWHW – 0.43 – for CeO2 – based phase, peak area – 1713, peak maximum – 13.459, height – 2086, FWHW – 0.58 – for ZrO2 – based phase. So area(CeO2)/area(ZrO2)=1.6. However, Zr and Ce atoms have different scattering factors. Rietveld refinement takes different scattering factors into account.
From the Rietveld method, the Ce0.5Zr0.5 catalyst contains 40 wt.% CeO2 and 60 wt.% ZrO2. Analysis of call parameters indicates that there are two solid solutions, for which the following designations are chosen: Cex1Zr1-x1O2 (40 wt.%) and t-Cex2Zr1-x2O2 (60 wt.%). Since in this case, both solid solutions consist only of Ce, Zr and O, it is possible to use the calibration curve (Vegard's rule) constructed from the lattice parameter for Ce-Zr solid solutions according to the literature data. This calculation showed the occupancy of Ce in the first solid solution x1 = 0.95, and in the second x2 = 0.24. This agrees with the initial ratio Ce/Zr = 1 during synthesis: after calculation Ce/Zr=0.34/0.37.
- Table 4. How the O/Me ration may be higher than 2 in mixed oxides?
Table 4 shows information about surface concentration of elements from XPS results (The title of table 4 has been changed). XPS gives information about the electronic properties and ratios between the elements on the surface of the catalysts. According to the literature data [Nelson, A.E.; Schulz, K.H. Surface chemistry and microstructural analysis of CexZr1−xO2−y model catalyst surfaces. Applied Surface Science 2003, 210, 206-221, doi:https://doi.org/10.1016/S0169-4332(03)00157-0.], CexZr1-xO2 is characterized by the surface ratio of O/(Ce+Zr)=2.3-2.7. These values exceed the stoichiometric value due to the enrichment of the surface with oxygen. For Ce0.5Zr0.5, this value was 2.4. In our work, for Mn0.1Ce0.45Zr0.45 and Mn0.2Ce0.4Zr0.4 catalysts, O/Me=2.31-2.35 (Table 4). According to the XRD data, with the increase in manganese content, crystalline oxides such as Mn2O3 and Mn3O4 are formed (Table 3), in which O/Mn<2. Therefore, the ratio of O/Me should decrease with an increase in the Mn content as it observed from XPS data (Table 4).
- Find the correlation between activity activity and structure for such mixed oxide catalysts is dificult. Probably the discussion of TPR data (for example in intensity of peaks at 200-300 oC) should be taken into account, because the main temperature of propane oxidation is in same region.
According to reviewer recommendation, the intensities of TPR peaks before 300C has been estimated. We used the profile integration before 300C. The following values were obtained: 0.35, 0.56, 0.68, 0.94, 1.37, 1.21 mmol (H2)/g with an increase in the Mn/(Ce+Zr+Mn) ratio from 0.1 to 0.6
The following sentences have been added in the description of TPR results and discussion. «It is worth mentioning, that the amount of hydrogen consumption before 300°C gradually grows from 0.35 to 1.37 mmol (H2)/g with an increase in the Mn/(Ce+Zr+Mn) ratio from 0.1 to 0.5. Further increase in the Mn/(Ce+Zr+Mn) ratio to 0.6 leads to decrease the amount of adsorbed H2 before 300°C to the value of 1.21 mmol (H2)/g. The amount of low-temperature hydrogen consumption correlates well with catalytic activity (Figure 5, Figure SI 1).» …«According to TPR data, low-temperature reducibility of catalyst correlates well with catalytic activit. (Figure 5, Figure SI 1).»
Reviewer 2 Report
This study is about the synthesis of Mn-Ce-Zr oxide catalysts and their application for CO and propane oxidation. this work is interesting but I have some suggestions for manuscript improvement:
- The Introduction should clearly illustrate the importance and novelty of the work, in this paper which is the hybrid material. Also need more references to prove this.
- Maybe the author should compare their results clearly with other reported works, highlighting the advantage and disadvantages of their novel composite.
- The manuscript contains some minor typo/grammar errors, please check all of it.
Author Response
This study is about the synthesis of Mn-Ce-Zr oxide catalysts and their application for CO and propane oxidation. this work is interesting but I have some suggestions for manuscript improvement:
- The Introduction should clearly illustrate the importance and novelty of the work, in this paper which is the hybrid material. Also need more references to prove this.
The authors agree with the remark. We emphasized the relevance and novelty of the work in the introduction section, and added links to recently published literature on the topic (pages 1-2).
- Maybe the author should compare their results clearly with other reported works, highlighting the advantage and disadvantages of their novel composite.
The authors agree with the remark. We compared the results obtained in this work with the results of other researchers for ternary oxide Mn-Zr-Ce catalysts in CO and propane oxidation reactions (Table 5)
- The manuscript contains some minor typo/grammar errors, please check all of it.
According to reviewer recommendation, we have checked typo/grammar errors
Reviewer 3 Report
In this manuscript, the performance of Mn-Zr-Ce catalyst prepared by coprecipitation method was studied, and the influence of composition on the catalyst, structure, microstructure and redox was also investigated. It is a meaningful manuscript, which can be used to study a tri- metal composite catalyst material for catalysis. However, some details need to be improved.
1. The obvious and serious problem in this manuscript is titled with the proportion of cations, but the proportion of metals in the full text is presented in the form of elements, e.g., manganese ions have multiple valence states. It is not appropriate to replace the original manuscript with manganese, and it is contrary to the title proposed. Please clarify the content further and revise it.
2. Details of the article need to be further checked and revised. For example, in the summary section, the performance index of nitrogen adsorption is described as the characterization of the catalyst, however the denominator of the metal proportion is not bracketed. Please further check and improve the details.
3. The XRD color of catalyst is not easy to distinguish and the content is not clear. It is suggested that the author modify the XRD characterization diagram.
4. At the end of 3.2, it is suggested that the increase of activity of ternary system compared with binary system may be related to the additional disorder of solid solution. There is no obvious evidence to show this phenomenon. Please further elaborate whether there are other reasons for this phenomenon.
Author Response
In this manuscript, the performance of Mn-Zr-Ce catalyst prepared by coprecipitation method was studied, and the influence of composition on the catalyst, structure, microstructure and redox was also investigated. It is a meaningful manuscript, which can be used to study a tri- metal composite catalyst material for catalysis. However, some details need to be improved.
- The obvious and serious problem in this manuscript is titled with the proportion of cations, but the proportion of metals in the full text is presented in the form of elements, e.g., manganese ions have multiple valence states. It is not appropriate to replace the original manuscript with manganese, and it is contrary to the title proposed. Please clarify the content further and revise it.
According to reviewer recommendation, the manuscript title has been changed to « The formation of Mn-Ce-Zr oxide catalysts for CO and propane oxidation: the role of element content ratio»
- Details of the article need to be further checked and revised. For example, in the summary section, the performance index of nitrogen adsorption is described as the characterization of the catalyst, however the denominator of the metal proportion is not bracketed. Please further check and improve the details.
According to reviewer recommendation, we have checked typos. Unfortunately, we did not understand the meaning of « the performance index of nitrogen adsorption is described as the characterization of the catalyst, however the denominator of the metal proportion is not bracketed. »
- The XRD color of catalyst is not easy to distinguish and the content is not clear. It is suggested that the author modify the XRD characterization diagram.
According to reviewer recommendation, Figures 2,4,6 have been modified.
- At the end of 3.2, it is suggested that the increase of activity of ternary system compared with binary system may be related to the additional disorder of solid solution. There is no obvious evidence to show this phenomenon. Please further elaborate whether there are other reasons for this phenomenon.
Indeed, disorder of solid solution is not correct definition in this context. To improve clarity of manuscript «disorder» was replaced by «restructuration». According to XRD data (Table 3), for ternary system the following phenomena are observed: a decrease in crystallite size of Ce-Zr-Mn oxides compared to binary systems, the formation of a second solid solution, and the formation of ternary solid solutions Ce(Mn,Zr)O2 and Zr(Ce,Mn)O2. These factors indicate additional restructuration of solid solution.
The following sentences have been added into the manuscript « It is likely that the increased activity of ternary systems compared to binary systems is associated with additional restructuration of the solid solution (formation of ternary Zr(Ce,Mn)O2 oxides, a decrease in crystallite size, appearance of second phase)»